# Enhancing Bone Cement Efficacy with Hydrogel Beads Synthesized by Droplet Microfluidics

**DOI:** 10.3390/nano14030302

**Published:** 2024-02-01

**Authors:** Zeyu Wang, Sherwin Yang, Chunjie He, Chaoqiang Li, Rong-Fuh Louh

**Affiliations:** 1Frontier Institute of Science and Technology (FIST), Micro- and Nano-Technology Research Center of State Key Laboratory for Manufacturing Systems Engineering, Xi’an Jiaotong University, Xi’an 710049, China; zeyu.wang@xjtu.edu.cn; 2Master’s Program of Biomedical Informatics and Biomedical Engineering, Feng Chia University, Taichung 407, Taiwan; 3Electronic Materials Research Laboratory, Key Laboratory of the Ministry of Education International Center for Dielectric Research & Shannxi Engineering Research Center of Advanced Energy Materials and Devices, Xi’an Jiaotong University, Xi’an 710049, China; hcj_0@stu.xjtu.edu.cn (C.H.); chaoqiangli@stu.xjtu.edu.cn (C.L.); 4Department of Materials Science and Engineering, Feng Chia University, Taichung 407, Taiwan

**Keywords:** microfluidics, droplet synthesis, bone healing, hydrogel drug delivery

## Abstract

Effective filling materials, typically bone cements, are essential for providing mechanical support during bone fracture treatment. A current challenge with bone cement lies in achieving continuous drug release and forming porous structures that facilitate cell migration and enhance osteoconductivity. We report a droplet microfluidics-based method for synthesizing uniform-sized gelatin hydrogel beads. A high hydrogel concentration and increased crosslinking levels were found to enhance drug loading as well as release performance. Consequently, the droplet microfluidic device was optimized in its design and fabrication to enable the stable generation of uniform-sized droplets from high-viscosity gelatin solutions. The size of the generated beads can be selectively controlled from 50 to 300 μm, featuring a high antibiotic loading capacity of up to 43% dry weight. They achieve continuous drug release lasting more than 300 h, ensuring sustained microbial inhibition with minimal cytotoxicity. Furthermore, the hydrogel beads are well suited for integration with calcium phosphate cement, maintaining structural integrity to form porous matrices and improve continuous drug release performance. The uniform size distribution of the beads, achieved through droplet microfluidic synthesis, ensures predictable drug release dynamics and a measurable impact on the mechanical properties of bone cements, positioning this technology as a promising enhancement to bone cement materials.

## 1. Introduction

The improper treatment of bone fractures often leads to prolonged recovery, diminished productivity, and potential disability. Healing a bone fracture is a complex physiological process that necessitates both biological and mechanical reconstruction [1]. Effective healing requires more than just the physical bridging of fractures; it also involves the migration and differentiation of cells within the fracture sites to facilitate tissue regeneration and self-renewal [2]. This process of cellular regeneration is crucial in endowing the tissue at the fracture sites with self-renewal capabilities, which are vital for preventing long-term secondary damage and unexpected side effects due to inflammation and structural degradation. A key challenge in fracture healing is replicating the extracellular matrix, which plays a crucial role in inducing tissue synthesis, especially in complex fractures that create physical gaps between bone fragments [1,3]. The current gold standard in the therapeutic process for non-healing defects is the use of bone autografts or allografts, attending the closest similarity in organic and inorganic composition to the damaged tissue. However, these materials are limited by availability and carry risks of potential disease transmission [4]. Additionally, successful fracture regeneration requires filling materials that retain biological functionalities and provide long-term mechanical support, stressing the need for ongoing research into suitable biomaterials such as bioactive ceramics, polymer composites, and bone cements, which can support tissue integration and healing while producing structural stability [3,5].

In the array of materials used for fracture healing, organic and inorganic bone cements stand out as superior choices, and are predominantly represented by poly(methyl methacrylate) (PMMA) and calcium phosphate cement (CPC), [3,6], respectively. These bone cements, prepared by mixing powdered and liquid phases, offer excellent injectability for effective fracture filling. Their hardening property not only assures mechanical support, but also prevents the material from leaking into adjacent tissues, thereby averting chronic irritation, inflammation, and potential subsequent surgical interventions [7,8]. Currently, CPC is favored over PMMA due to its attractive characteristics such as its higher biocompatibility [9], osteoconductivity [10], biodegradability, drug delivery capabilities [11], and lower exothermic reaction during the setting process, which minimizes the risk of thermal damage [12]. However, the drug delivery potential of CPC is limited by its incorporation methods. Standard practices, such as the straightforward mixing or coating of drug molecules with CPC particles, would face limitations in loading capacity and tend to exhibit rapid release [13,14,15]. The purposes of drug incorporation are preventing infection and encouraging tissue regeneration, both of which necessitate a controlled and sustained release profile [16]. Additionally, the escalation of the pore sizes in CPC scaffolds is critical for facilitating cell migration, thus enhancing cell infiltration and regeneration ability [17,18]. To advance drug delivery and cellular penetration, hydrogel porogens have been introduced into the CPC composition [19,20]. These polymeric beads not only increase drug loading capacity, but also degrade to form larger pores. Unfortunately, the variability in bead sizes results in inconsistent drug release rates and poses challenges to the cement scaffold’s mechanical stability [21].

In this study, we present a microfluidic methodology for fabricating gelatin hydrogel beads, aimed at achieving prolonged drug release and enhanced cell infiltration. Microfluidic droplet generation systems are extensively utilized in applications such as single-cell analysis [22], drug screening [23], micro-reaction system construction [24,25], and drug carrier synthesis [26]. These systems produce a dispersed phase, or droplet, within a continuous phase that exhibits contrasting hydrophilic or hydrophobic properties relative to the dispersed phase [27]. Droplet formation in microfluidics is typically governed by shear forces and surface tension interactions at microchannel junctions, where immiscible fluids intersect or flow coaxially, facilitating the controlled segmentation of the dispersed phase into uniform, discrete droplets [28,29,30]. Under stable conditions, including consistent flow injection rates, fluid viscosity, and device geometry, these systems can reliably and rapidly produce droplets of uniform volume. However, a significant challenge in synthesizing hydrogel beads using microfluidic droplet generation is managing the high viscosity of pre-crosslinked polymer solutions [31,32]. This high viscosity not only complicates droplet formation, but also exacerbates mixing difficulties, a common issue in droplet microfluidics [33,34]. Traditional droplet microfluidics-based hydrogel bead synthesis encounters problems such as low crosslinking rates, extended incubation times, and variations in bead performance due to non-uniform crosslinking [35,36]. By refining channel fabrication techniques and reagent formulations, we have successfully generated droplets from gelatin solutions premixed with crosslinkers, achieving a uniform crosslinking rate throughout the beads. Unlike beads produced by traditional mixing methods, those synthesized via droplet microfluidics exhibit uniformity in size, contributing to consistent drug release kinetics and mechanical stability. Systematic enhancements in the performance of these hydrogel beads within calcium phosphate cement (CPC) have been realized through advancements in microfluidic droplet generation technology, encompassing device fabrication, surface modification, and material selection. The droplet generation apparatus was constructed using micro-milling of poly(methyl methacrylate) (PMMA) plates, offering greater flexibility and reduced costs compared to the conventional photolithography technique, which involves multiple processing steps and an expensive photoresist and substrates. The PMMA microfluidic channels underwent plasma treatment to yield hydrophobic surfaces, preventing gelatin adsorption and subsequent channel obstruction, thereby prolonging device durability. Achieving optimal drug loading and sustained release from gelatin hydrogels usually need a high concentration of the precursor solution for dense polymer network formation. However, increased gelatin concentration resulted in augmented viscosity, which is a critical step for single emulsion hydrogel droplet formation to impede the droplet breakup. The introduction of Na^+^ ions into the gelatin solution effectively reduced its viscosity, and the hydrophobic channel treatment mitigated gelatin adhesion at low temperatures.

With the implemented modifications, the droplet microfluidic system consistently produced gelatin beads ranging from 50 to 300 μm, utilizing a solution containing a premixed precursor and crosslinker. This system’s remarkable tolerance for high-viscosity liquids enabled the gelatin concentration to reach 0.04 g/mL at room temperature, offering greater ease of handling compared to the lower concentration of 0.01 g/mL or the requirement for heating to 50 °C in organic solvents [37,38], which inhibit the utilization of cells or bioactivity reagents including some antibiotics and cytokines. The synthesized gelatin beads exhibited an enhanced loading capacity for vancomycin, an antibiotic effective against Staphylococcus aureus infections typically encountered during bone cement application. Notably, the maximum dry weight ratio of vancomycin in the beads exceeded 40%, which is significantly higher than the conventional loading capacity, usually less than 20% [39,40]. This facilitates the prolonged release and effective inhibition of microbial proliferation. Additionally, an increase in the concentration of the crosslinker was found to moderate the vancomycin release kinetics, resulting in a more gradual release at the initial stage and thereby mitigating the cytotoxic effects associated with burst drug release. When integrated with injectable Calcium Phosphate Cement (CPC), these beads maintained their stability and significantly enhanced the drug release profile of the biphasic cement. The bead–cement complex also exhibited improved drug release kinetics, which was attributed to the microfluidic-synthesized beads. Furthermore, these beads contributed to the formation of pore structures within the cement, serving as ideal channels for cell migration and thus bolstering osteoconductivity.

## 2. Materials and Methods

### 2.1. Microfluidic Chip Fabrication and Droplet Synthesis

The microfluidic channel was designed by SolidWorks (Dassault Systems, Vélizy-Villacoublay, France). The designed pattern was engraved on a PMMA plate (Sigma-Alderich, St. Louis, MO, USA) through a Micro-Engraving System (Roland, Osaka, Japan). The engraved PMMA plate was sealed with another PMMA plate by heat annealing (Annealsys, Montpellier, France). The microfluidic chip was then coated with Tetramethyldisiloxane (Lancaster Synthesis Inc., Ward Hill, MA, USA) through plasma-enhanced chemical vapor deposition (Sky Technology, Shenyang, China) for 15 min with 200 w coating. Gelatin Type A 300 Bloom (Sigma-Alderich, St. Louis, MO, USA) was premixed with glutaraldehyde (Sigma-Alderich, St. Louis, MO, USA) in a phosphate-buffered saline powder solution (PBS) as the dispersed phase, and liquid paraffin (Sigma-Alderich, St. Louis, MO, USA) containing 0.5% of span 80 surfactant (Sigma-Alderich, St. Louis, MO, USA) was used as the continuous phase. Vancomycin (Sigma-Alderich, St. Louis, MO, USA) was premixed with the dispersed phase when required. Both the dispersed phase and continuous phase were pumped by a syringe pump (Tricontinent Scientific Inc., Auburn, CA, USA). The gelatin beads were washed by ether and dried by a vacuum freeze dryer (Tuohe, Shanghai, China).

### 2.2. Hydrogel Bead Validation and Drug Release Evaluation

To evaluate the crosslinking level of the gelatin, the beads were validated by Ninhydrin assay (Sigma-Alderich, St. Louis, MO, USA) by a 100 °C water bath for 20 min. The free amine amount was evaluated by 570 nm absorption using UV/visible spectroscopy (Shimadzu, Kyoto, Japan). The drug loading capacity was evaluated by grinding weighted vancomycin-loaded gelatin bead samples. Drug releasing was achieved by the continuous shaking of bead debris in 400 mL of PBS using an orbital shaker (Benchmark Scientific, Sayreville, NJ, USA) for 2 h. The supernatant was collected and evaluated by 280 nm absorption using UV/visible spectroscopy (Shimadzu, Kyoto, Japan). The evaluation of drug releasing kinetics follows a similar process without grinding. Similar weight vancomycin-loaded gelatin bead samples were incubated in the 96-well culture plates with 200 μL of PBS added in a 37 °C incubator for monitoring the in vivo environment and evaluated by spectroscopy by fixed time points.

### 2.3. Bacterial Culture and Proliferation Validation

Staphylococcus aureus was cultured in a Lysogeny broth (LB) culture medium at 37 °C. Bacteria were collected and diluted to McFland 0.5, referring to a standard concentration of 1.5 × 10^8^ CFU/mL. Vancomycin-loaded gelatin beads containing a total of 2.15 mg of vancomycin were suspended in 17.2 mL of LB culture medium and diluted to a final concentration of 128, 64, 32, 16, 8, 4, 2, 1, 0.5, and 0.25 μg/mL of suspension medium. The bacteria solution was diluted in this bead suspension medium to form a concentration of 1.5 × 10^6^ CFU/mL. The control groups were prepared by free vancomycin LB solutions with similar concentrations. Both the experimental and control groups were incubated at 37 °C for 24 h. OD600 was evaluated by UV/visible spectroscopy (Shimadzu, Kyoto, Japan) before and after incubation.

### 2.4. Cell Culture and Viability Validation

Human chondrocytes were obtained from the hospital with the donator informed. Cells were cultured in a 96-well plate in a 37 °C incubator with 5% CO_2_ concentration. After the cell density reaches 50% of the well area, the culture medium containing 128 μg/mL of the final concentration of vancomycin through gelatin beads was used for replacing the original culture medium. The MTT assay (Thermo Fisher, Waltham, MA, USA) was processed after 24 h of culture.

### 2.5. Creating and Validating Gelatin Bead–CPC Cement Complex

The vancomycin-loaded gelatin beads were first mixed with the liquid phase of a commercial CPC bone cement, Surgical Simplex P (Stryker, Kalamazoo, MI, USA). The liquid phase containing the gelatin beads was then mixed with powder components to achieve a final weight ratio between the beads and cement of 1:4. After thorough mixing, the gelatin bead–CPC complex was incubated at room temperature for 30 min to solidify. For evaluating drug release kinetics, 5 g of the complex was incubated in water, compared to 1 g of the vancomycin-loaded gelatin beads. After 10 min of a 90 °C water bath for dissolving the gelatin beads, the inner surface morphology of the gelatin bead–CPC complex was evaluated using a field emission scanning electron microscope (FE-SEM) S-4800 (Hitachi, Chiyoda City, Japan). Compressive strength testing was performed with a 50-ton micro-computer universal mechanical tester, Model No. CY-6040A1 (Taichung, Taiwan). The control group for morphological and mechanical evaluation was formed using CPC cements that did not contain gelatin beads.

## 3. Results

### 3.1. Device Design and Droplet Generation

The microfluidic system was designed to utilize a flow-focusing droplet generation mechanism (Figure 1). Unlike the T-junction and co-flow devices, the flow-focusing design facilitates droplet breakup through the stretching of the dispersed phase, driven by increased flow velocity and the shear stress exerted by the continuous phase [30,41,42]. This mechanism allows the generation of smaller droplets and provides better adaptability for high-viscosity fluids. In contrast to the T-junction system, flow focusing effectively prevents gelatin solution adherence to channel walls and subsequent obstruction, thanks to the focusing effect created by the dual continuous phase inlets [30]. Moreover, in flow-focusing devices, droplet size is predominantly influenced by flow rates rather than geometric dimensions, suggesting that wider channels are capable of producing smaller droplets. While the droplet sizes in the T-junction and co-flow systems depend heavily on the channel width and capillary diameter, the flow rate-controlled sizing in flow focusing expands fabrication tolerance and enables the use of micro-milling on PMMA for cost effectiveness. The effectiveness of Tetramethyldisiloxane (TMDSO) hydrophobic treatment in reducing the affinity between gelatin solution and PMMA is evidenced by the decreased contact angle as Figure 1 shows. With these optimizations, a PMMA channel measuring 4.5 mm in width and 320 μm in depth is capable of generating micrometer-sized gelatin droplets. To avoid the fluctuation of the ratio between the crosslinking agent and precursor, the gelatin solution was premixed with a different concentration of glutaraldehyde. The crosslinking degrees of the gelatin beads were evaluated by the quantification of residual free amine, which demonstrated that a 22.5% crosslinking rate of amine could be achieved by 2.5 mM of glutaraldehyde. Although the crosslinking rate shows the increasing tendency with glutaraldehyde concentrations, to avoid rapid hydrogel fixation and blocking channel glutaraldehyde, a concentration higher than 2.5 mM was not used.

### 3.2. Drug Loading Capacity and Releasing Kinetics of Hydrogel Beads

Vancomycin, known for its broad-spectrum antibacterial properties, was chosen as the drug for loading. The drug loading capacity was initially analyzed across gelatin particles of varying sizes and varying amounts of antibiotics (Figure 2a), all produced from a 25 mL solution containing 5 to 20 mg of vancomycin, 1 g of gelatin, and 0.5 mM of glutaraldehyde in the final concentration. The capacity was assessed by the dry weight ratio of vancomycin to the beads produced. Among the 50, 100, 200, and 300 μm gelatin beads, the 50 μm beads exhibited a significantly lower loading capacity. In contrast, the three larger sizes demonstrated capacities exceeding 40%, with no marked differences among them, suggesting that increasing the bead size beyond a certain point does not proportionally enhance the drug loading capacity. Given that larger hydrogel beads create larger pores in cements, potentially diminishing mechanical strength, 200 μm beads were selected for subsequent evaluations to balance mechanical integrity and the ease of cell migration. Further analysis was conducted on the impact of varying vancomycin concentrations (5, 10, 15, 20 mg) in the gelatin solution. While higher vancomycin levels increased the overall drug loading capacity, the proportion of the drug loaded into the gelatin droplets showed a decline (Figure 2b). The drug release dynamics of these gelatin beads were assessed in a water curing environment for 480 h (Figure 2c). The results indicated that higher drug loading capacities and lower loading ratios led to an initial burst release. Considering that a reduced loading ratio could result in the excessive early release of free vancomycin, potentially toxic to tissues, maximizing drug loading could pose risks to the system’s toxicity. An additional experiment increasing the crosslinking agent concentration to 1.5 and 2.5 mM demonstrated an improved performance in terms of a slower drug release (Figure 2d), suggesting that higher crosslinking rates can enhance drug release profiles and mitigate the burst release.

### 3.3. Antibiotic Efficiency and Tissue Toxicity of the Beads

To assess the antibiotic efficacy of vancomycin-loaded gelatin beads, a Staphylococcus aureus culture with a 1.5 × 10^5^ CFU/mL standard concentration was utilized to create a vancomycin environment ranging from 0.25 to 128 μg/mL (Figure 3a). Vancomycin concentration was estimated based on the drug loading capacity of the 200 μm gelatin beads, and a corresponding free vancomycin solution served as the control. Bacterial proliferation was measured by OD600, starting at 0.008. After 24 h, the vancomycin-free solution reached an OD600 of 0.586. Notably, only the 0.25 μg/mL vancomycin bead group exhibited a significant increase in OD600 to 0.516, indicating the ineffective inhibition of bacterial growth. Conversely, the 0.5 μg/mL vancomycin bead group showed a reduced OD600 increase of 0.131, and groups with higher concentrations exhibited negligible OD600 changes, suggesting successful bacterial growth inhibition. Interestingly, in the free vancomycin groups, concentrations of 0.25, 0.5, and 1 μg/mL failed to inhibit bacterial proliferation effectively, as evidenced by the final OD600 values of 0.551, 0.435, and 0.274, respectively. This discrepancy in antibacterial performance could be attributed to the sustained release of vancomycin from the beads, which more effectively maintains continuous bacterial growth inhibition. Chondrocytes are commonly used for evaluating the biocompatibility of bone cements in the healing of osteochondral defects and their effectiveness in cartilage repair [43,44]. While this cell type does not directly assess the osteoconductivity of bone cements, it offers more accessible approaches to evaluate biocompatibility. The cytotoxicity of the vancomycin-loaded gelatin beads was further evaluated by 12, 24, and 48 h of chondrocyte culture followed by a viability assessment using the methylthiazolyldiphenyl-tetrazolium (MTT) assay. The estimated vancomycin concentrations showed no adverse impact on cell viability. Notably, beads pre-coated with 2% of BSA solution exhibited a higher viability of 93.5% compared to the non-coated beads at 86.2% (Figure 3b), indicating a low toxicity. Notably, the live/dead cell staining results indicate that both BSA-incubated and non-incubated beads do not cause a marked increase in cell death compared to the control group (Figure 3c). Given the absence of notable cytotoxicity, the observed differences in cell viability among the three groups could be attributed to the vancomycin-induced suppression of cell proliferation.

### 3.4. Bone Cement Loading Ability

Vancomycin-loaded 200 μm gelatin beads were incorporated into unset Calcium Phosphate Cement (CPC) at a 1:1 volume ratio. The formation of pore structures in the solidified cement was observed, suggesting the hydrogel beads’ resilience to the exothermic hardening process of the cement (Figure 4a). The drug release kinetics of the gelatin-enhanced CPC were assessed over 480 h in a water culture, demonstrating a sustained release lasting to the end of the test. This indicates that the hybrid cement facilitates prolonged drug release, potentially providing an extended microbial inhibition crucial for successful wound healing and tissue regeneration. Notably, the hybrid scaffold, comprising both hydrogel beads and CPC, exhibited elongated and more consistent drug release kinetics compared to the beads alone. This variation is likely attributable to the hydroxyapatite in the CPC, which offers additional surface adsorption and ionic interactions, enhancing drug affinity. Compared to CPC cements without added beads, the 200 μm microfluidic-synthesized gelatin beads create uniform-sized pores nearly identical to their size in the cement matrix (Figure 4c). Interestingly, while pure CPC cements cracked under compressive strength ranging from 67.95 to 73.16 MPa, the gelatin bead-enhanced CPC cements exhibited improved compressive strength resistance, cracking only at strengths exceeding 118.9 Mpa. However, this enhanced mechanical performance might be attributed to the increased flexibility provided by the gelatin beads. Although not quantified, it was observed that CPC cements with gelatin beads demonstrated greater deformation before cracking compared to pure CPC cements, indicating that the complex is more ductile and resilient, whereas pure CPC cement is more brittle. Given that bone cements are required to offer mechanical support, the therapeutic implications of higher compressive strength resistance coupled with greater deformation in the gelatin bead-enhanced CPC cements remain uncertain. Despite this, the smooth drug release kinetics and the presence of pore structures suggest that the integration of gelatin beads into CPC significantly improves its osteoconductivity and drug release capabilities, providing both cell migration channels and a continuous drug release mechanism.

## 4. Discussion

In this study, the development of gelatin beads using droplet microfluidics emerges as a pivotal innovation, enhancing drug release kinetics and cell infiltration efficacy in Calcium Phosphate Cement (CPC) bone cements. The traditional hydrogel particles in drug delivery are synthesized through mixing, yielding a wide range of particle sizes [20,45]. This variability is significant since drug loading capacity and release kinetics are closely tied to particle size. Mixed synthesis often leads to unpredictable release curves and the risk of a burst releasing large drug quantities. Such variability not only induces cytotoxicity due to the high early-stage drug release, but also creates unpredictable dynamic influences in the cement matrix [46]. Varied-sized hydrogel particles result in inconsistent porosity [47]; overly small pores hinder cell migration and osteoconductivity [18], while excessively large ones weaken the cement’s support and may cause premature biodegradation before tissue regeneration [48]. Therefore, the introduction of hydrogel into bone cements is a balance between retaining structural integrity and gaining additional benefits like drug release, osteoconductivity, and controlled biodegradation.

Compared with mixing approaches, droplet microfluidics advance by their unique capability to rapidly generate droplets of uniform size. However, this method encounters challenges due to the sensitivity of microfluidic channels to clogging. The issue is exacerbated when processing solutions with high polymer concentrations, as their increased viscosity hinders intra-droplet mixing [49]. Traditional droplet microfluidic systems, which rely on the hydrodynamic effects created by curved channels for mixing [30,33], struggle with high-viscosity solutions that reduce hydrodynamic intra-droplet streaming, leading to failures in mixing. This mixing between the polymer and crosslinker, a necessary step in forming hydrogel, complicates the use of chemical, ionic, pH, and enzymatic crosslinking methods [50]. Current approaches in droplet microfluidic hydrogel particle synthesis typically involve lowering the concentration of the polymer or crosslinker to improve mixing [51], or using photo-initiators that enable polymerization through optical stimulation [52]. However, lower concentrations of reagents can affect the durability of the hydrogel and accelerate drug release, while photo-initiator based hydrogels pose concerns of cost and biocompatibility due to the toxic byproducts from the light-induced crosslinking process. These drawbacks make such methods less ideal for application in bone cement materials.

Diverging from traditional droplet microfluidics approaches for hydrogel particle synthesis, this study employs a method based on chemical crosslinking with a high concentration of gelatin, noted for its low toxicity and excellent biocompatibility. We addressed the common issue of mixing by premixing gelatin and the crosslinker, which, while posing a risk of channel clogging, led to significant advancements in the channel de-sign. These improvements include wider channels and a hydrophobic coating, culminating in a device that consistently produces uniform-sized gelatin beads. Notably, these beads exhibit a high drug loading capacity, surpassing 40%, which significantly exceeds the conventional capacity, which is typically less than 20% [39,40]. Crucially, this enhanced loading capacity does not result in an early-stage burst release, thanks to the increased amount of the crosslinker. The extended drug release duration, lasting up to 200 h without an initial burst phase, represents a substantial leap in drug delivery systems. This controlled drug release is particularly significant as it effectively tackles a prevalent challenge in bone repair treatments, which is the need for sustained and consistent drug delivery without overwhelming the surrounding tissues.

Furthermore, the integration of these vancomycin-loaded gelatin beads with CPC bone cement represents a significant stride in orthopedic applications. The beads not only maintained their structural integrity within the cement, but also created pores that potentially facilitate cell migration and tissue integration. This dual functionality of enhancing drug release while simultaneously reinforcing the mechanical properties of the CPC cement is particularly promising. The study’s findings on the minimal cytotoxicity of the beads add another layer of value, confirming their biocompatibility and suitability for medical applications. Given these advancements in both the fabrication of uniform size beads with improved drug release kinetics and the multifunctional role of these beads in CPC cements, the study presents a compelling case for the potential of microfluidic-generated gelatin beads in revolutionizing CPC cement technology for bone repair and beyond.

## 5. Conclusions

In summary, we have developed gelatin beads using droplet microfluidics to enhance the drug release kinetics and cell infiltration efficacy of CPC bone cements. The microfluidic channel, fabricated from PMMA via mechanical etching, offers cost-effective advantages. The incorporation of flow-focusing droplet generation enables the use of larger channel dimensions, synergizing with hydrophobic treatment to facilitate the flow of high-viscosity gelatin solutions without channel blockage. This microfluidic system also allows for the premixing of gelatin crosslinkers, achieving a heightened drug loading capacity and consistent drug release profiles. We attained a drug loading capacity exceeding 40%, meaning that more than 40% of the dry weight of the formed gelatin beads comprised the loaded drug, which is applicable to beads with diameters of 100, 200, and 300 μm. The drug release duration extended to 200 h, characterized by the absence of an initial burst release phase. The vancomycin-loaded gelatin beads exhibited effective antibiotic properties and minimal cytotoxicity, confirming their biocompatibility and infection prevention capabilities. Additionally, these beads integrated seamlessly with CPC cement, maintaining structural integrity post-solidification and creating pores within the cement matrix. The resultant hybrid CPC cements displayed a sustained drug release profile, affirming the successful incorporation of gelatin beads. The size of the fabricated beads, controllable within the range of 50 to 300 μm, is suitable for cell migration, and their size uniformity predictably influences the mechanical properties of the CPC cements. Given these multifaceted enhancements in device fabrication, bead synthesis, infection prevention, and osteoconductivity, our droplet microfluidic-generated gelatin beads present a significant advancement in CPC cement technology.

## 6. Patents

This manuscript results have a patent PCN230014715 approved by the National Intellectual Property Administration, PRC.

## Figures and Tables

**Figure 1 nanomaterials-14-00302-f001:**
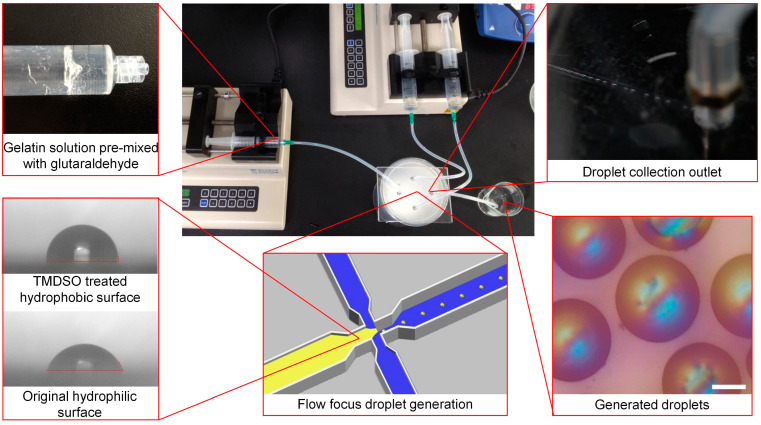
Microfluidic-based droplet generation. A flow self-focusing droplet generation channel was made by PMMA, gelatin solution premixed with crosslinker glutaraldehyde was pumped as the dispersed phase in the center inlet, focused by the two continuous phases from the two side inlets for generating droplets. The channel was hydrophobic coated with TMDSO for avoiding gelatin solution affinity with channel wall and blocking. Scale bar: 100 μm.

**Figure 2 nanomaterials-14-00302-f002:**
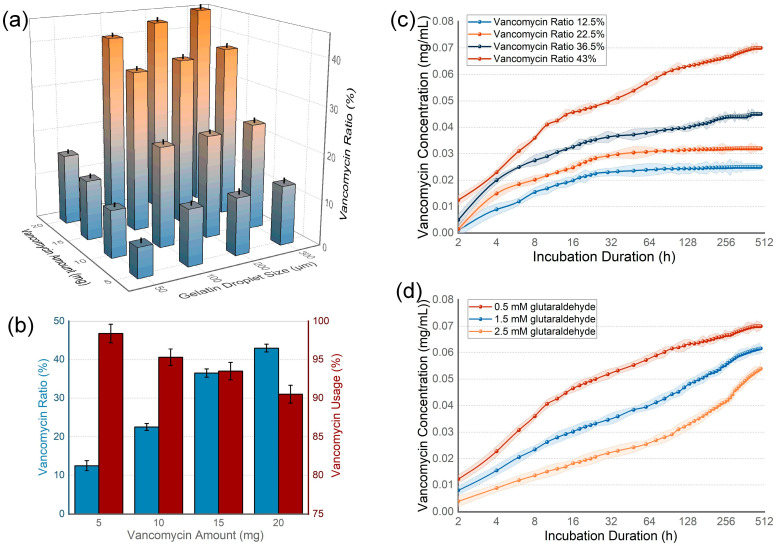
Drug loading capacity and drug releasing kinetics analysis of the gelatin beads. (**a**) Larger beads show higher drug loading capacity, but size increase after 100 μm demonstrates lower efficiency on loading capacity improvement. (**b**) Increased drug concentration improves the drug loading capacity, but causes lower loading rate that can induce drug waste. (**c**) Drug release kinetics influenced by ratio of loaded vancomycin, a higher ratio of vancomycin in the gelatin beads induces longer releasing period, but dramatic burst release in early stage. Error bars are demonstrated by the light color region surrounding the line. (**d**) Increased crosslinker rate concentration improves the drug release kinetics by lowering early stage releasing rate. Error bars are demonstrated by the light color region surrounding the line.

**Figure 3 nanomaterials-14-00302-f003:**
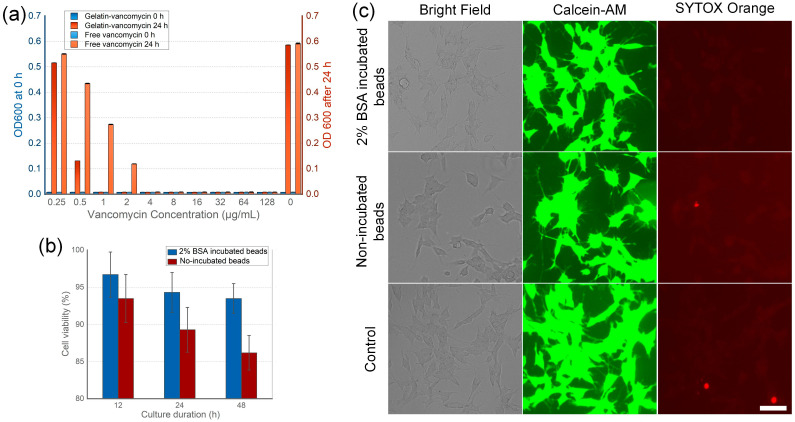
Biofunction of the vancomycin-loaded beads. (**a**) Compared with free vancomycin, vancomycin-loaded bead shows better microbe inhibition in lower concentrations. (**b**) Vancomycin-loaded beads demonstrate lower toxicity for chondrocyte, BSA coating can further improve the low cytotoxicity. (**c**) Chondrocytes showed high viability with low cell death rate evaluated by Calcein-AM and SYTOX Orange staining. Scale bar: 50 μm.

**Figure 4 nanomaterials-14-00302-f004:**
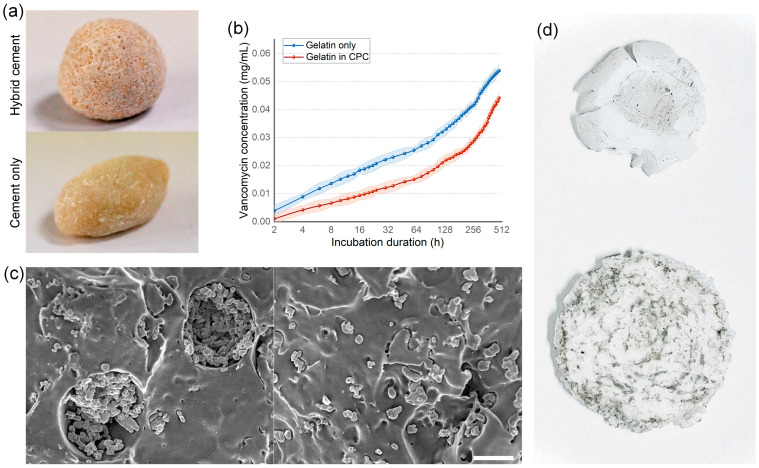
Gelatin beads show good performance when mixing with CPC cements. (**a**) The hybrid CPC cements containing gelatin beads demonstrate pore structure after solidification. (**b**) The hybrid cements show continuous drug releasing ability better than gelatin beads; only situation, error bars are demonstrated by the light color region surrounding the line. (**c**) SEM images show gelatin beads added CPC cement (**left**) containing pore structures approximate to 200 μm, while non-pore structures could be found in pure CPC cement (**right**). (**d**) In the compressive strength resistance test, pure CPC cement (**above**) shows a clear sign of crack in 67.95 to 73.16 MPa, while gelatin beads added CPC cement (**bottom**) show deformation first and cracks after compressive strength reaches 118.9 MPa. Scale bar: 100 μm.

## Data Availability

The original data are available from the corresponding authors upon reasonable request.

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
