# Peer review of "Enhancing Bone Cement Efficacy with Hydrogel Beads Synthesized by Droplet Microfluidics"

_nanomaterials, 2024, doi:10.3390/nano14030302_

Round 1

Reviewer 1 Report

Comments and Suggestions for Authors

The study of Wang et al. investigates the enhancement of bone cement efficacy using hydrogel beads synthesized via droplet microfluidics. While the article's structure is concise in scope and content, the figures exhibit good quality, and the English grammar is of publishable standard. However, in terms of materials characterization, essential techniques like SEM, XRD, and mechanical properties are absent. The addition of about details would enhance the understanding of the connections between structure and final properties. Below I am sending some specific comments to the manuscript.

Abstract

1) The abstract contains excessive general information. I recommend incorporating specific results achieved in this work to enhance clarity and provide a more focused overview.

Introduction

2) page 2; row 52: inorganic bone cements, predominantly poly(methyl methacrylate) (PMMA) and calcium phosphate cement (CPC), emerge as superior choices.- Please note that PMMA is organic and not inorganic cement.

3) The Intro part properly described the literature review of the given topic, however the novelty and the clear contribution of this study should be better highlighted.

2. Materials and Methods

4) Information regarding the type and preparation process of CPC should be added.

3. Results

5) page 5; row 205: Given that larger hydrogel beads create larger pores in cements, potentially diminishing mechanical strength...Although the authors mention mechanical properties, but none were tested in the article.

6) SEM observation will be useful to monitor the formation of pore structures in the solidified cement

Comments on the Quality of English Language

Author Response

Please see the attached PDF document.

Reviewer 2 Report

Comments and Suggestions for Authors

I read the article titled "Enhancing bone cement efficacy with hydrogel beads synthesized by droplet microfluidics" with interest. The article discusses a novel bone carrier based on hydrogel beads obtained through droplet microfluidics-based methods. The work is quite intriguing, and its main advantage lies in the authors' success in obtaining homogeneous beads and interesting release profiles both without and with the addition of calcium-phosphate cements.

However, while reading this manuscript, I have a few comments and questions:

  1. The process of combining beads with cement should be described in the Materials and Methods section.
  2. I did not find a description in the paper regarding the type of calcium-phosphate cement (CPC) used.
  3. Why were chondrocytes specifically chosen for the biological studies?
  4. What was the porosity of the obtained materials?
  5. For vancomycin release, there is a lack of error bars on the graph.

I would appreciate answers to the above questions.

Author Response

Please see the attached PDF document.

Round 2

Reviewer 1 Report

Comments and Suggestions for Authors

The authors have addressed all my concerns, therefore I recommend to publish this manuscript in the present form.

Reviewer 2 Report

Comments and Suggestions for Authors

The Authors have corrected the manuscript according to the suggestions.